# Microencapsulation of Cannabidiol in Liposomes as Coating for Cellulose for Potential Advanced Sanitary Material

**Julija Volmajer Valh** [1,*], **Zdenka Peršin** [1], **Bojana Vončina** [1], **Kaja Vrezner** [1], **Lidija Tušek** [2] **and Lidija Fras Zemljič** [1]

[1] Institute of Engineering Materials and Design, Faculty of Mechanical Engineering, University of Maribor, Smetanova 17, SI-2000 Maribor, Slovenia; zdenkapersin@gmail.com (Z.P.); bojana.voncina@um.si (B.V.); kaja.vrezner@gmail.com (K.V.); lidija.fras@um.si (L.F.Z.)

[2] Scientific Research Centre Bistra Ptuj, Slovenski trg 6, SI-2250 Ptuj, Slovenia; lidija.tusek@bistra.si

\* Correspondence: julija.volmajer@um.si; Tel.: +386-2-220-7897

**Abstract:** The microencapsulation of the cannabidiol and its integration into the tampon can eliminate vaginal inflammation, which at the same time lead to relaxation of the abdominal muscles. The tampon, which contains the active substance cannabidiol (CBD), was developed as an advanced fibrous composite for sanitary application. The active substances were microencapsulated, and, as a carrier, liposomes micro/nano capsules were used. The CBD liposome formulation was analyzed by particle size, polydispersity index, zeta potential, and encapsulation efficiency. Particle size of the CBD liposome liquid formulation was increased by 19%, compared to the liposome liquid formulation and the encapsulation efficiency of CBD in liposome particles, which was 90%. The CBD liposome formulation was applied to cellulose material. The composition of the fibrous composite material was evaluated by Fourier transform infrared spectroscopy, the fiber morphology was analyzed by scanning electron spectroscopy, while the bioactive properties were assessed by antioxidant efficiency, antimicrobial properties, and desorption kinetics. CBD liposome functionalized tampons have both antioxidant and antimicrobial properties. Antimicrobial properties were more pronounced against Gram-positive bacteria. The desorption kinetics of the CBD liposome immobilized on the surface of the composite material was studied using antioxidant activity in the desorption bath. The prepared CBD liposome functionalized tampon additionally shows higher biodegradability compared to references. This high-quality, biodegradable sanitary material based on microencapsulated CBD components as a functional coating provides a platform for many different applications besides medical textiles, also for packaging, pharmaceuticals, paper and wood-based materials, etc.

**Keywords:** microencapsulation; cannabidiol; liposome; coatings; tampon

## 1. Introduction

More than half of women suffer from monthly pain during their menstrual cycle. The pain occurs in the form of cramps, which are located in the lower part of the stomach and loins. In some cases, the pain is also accompanied by nausea, vomiting, headaches, diarrhea, altered cravings for food and/or sweating. The pain is caused by the contraction of the uterus, the reduced blood supply to the lining of the uterus, and the increased formation and release of inflammatory substances, so-called prostaglandins (PGs) in the inner layer of the uterus [1]. During the menstrual cycle, the pH value in the vagina also changes in many cases, which can lead to inflammation of the vagina [2,3]. To relieve the problems mentioned above, drugs are usually given in the form of tablets, capsules, or tinctures [1].

The synthetic drugs often cause side effects [4] that one wishes to avoid. In addition, there is also an issue of drugs accumulating after they are not needed anymore or past their case expiration date [5]. Accumulated unused drugs are classified as hazardous waste and should be recycled in specialized plants, following the highest standards once the final product is no longer useful as medicine and making sure it does not pose a risk to the

environment. Emphasizing the latter, substances of natural origin possessing the same or even greater effects should be disposed of following the same standards.

The cannabis plant has been cultivated and used since ancient times for its industrial benefits and medicinal applications, while more recently it has gained in attractiveness for the relief of menstrual pain [5]. The plant contains more than 80 different chemicals known as cannabinoids. The most common cannabinoid, tetrahydrocannabinol (THC), is known for its psychoactive properties, while cannabidiol (CBD) is the second most common ingredient [5] and has great potential as an analgesic, anti-epileptic, muscle relaxant, anxiolytic, neuroprotectant, and antipsychotic, as well as an anti-inflammatory [6–9], antioxidant [10,11] treatment for cancer-related pain [12]. CBD also affects some non-cannabidiol receptor systems in the brain and interacts with receptors that respond to a variety of drugs and neurotransmitters [6,7]. CBD can transmit messages to nerve cells, including opioid receptors, which are known for their role in the perception of little or great pain. The opioid receptors serve as painkillers such as heroin, fentanyl and morphine [13–16]. Because CBD is known for its anti-inflammatory properties, it may help to regulate pain caused by prostaglandins, the hormones that trigger uterine contractions in the first place. Further, CBD also has "relaxing and analgesic effects," i.e., it can induce the muscles in the abdominal cavity to relax and stop contracting so aggressively [17], which leads to dysmenorrhea.

Nevertheless, although CBD has multiple pharmacological actions, little is known about its safety and side effect profile in animals and humans. There is an abundance of public comment on the therapeutic effects of cannabinoids, indicating that there have been only a limited number of clinical studies on this topic due to the illegal status of cannabinoids in most countries [9]. Bergamaschi et al [18] reviewed in vivo and in vitro reports on CBD administration across a wide range of concentrations, based on reports retrieved from the Web of Science, SciELO, and Medline. It was reported that, based on advances in CBD administration in humans, controlled use of CBD may be safe in humans and animals. CBD use also carries some risks and causes some side effects. In general, the authors concluded that controlled administration of CBD may be safe in humans and animals. A more recent study on adverse effects and toxicity reported that CBD has proven therapeutic efficacy for serious conditions and is likely to be recommended off label by physicians for several conditions. However, adverse effects and potential drug–drug interactions must be taken into consideration by clinicians prior to recommending off label CBD [19].

Liposomes are among the most studied, used, and commercially approved systems for delivering drugs, nutrients, and/or genetic material into the human body [20]. Liposomes are spherical vesicles composed of a phospholipid bilayer that contains an interior aqueous core, providing space to encapsulate hydrosoluble substances [21]. Liposomes can trap both hydrophobic and hydrophilic compounds, prevent degradation of the trapped compounds, and release the trapped substances at specific targets. Liposomes allow CBD to be introduced into the human body orally, allowing precise dosing, with local (dermal) or systemic (transdermal) effects achieved locally (dermal) or systemically (transdermal) [22–24]. Although it is the first nanomedical delivery system to make the transition from concept to clinical application and is now an established technology platform with considerable clinical acceptance, CBD encapsulation into liposomes is not yet very recommended. The reason for this is that, during the encapsulation of lipophilic active compounds such as CBDs, they have a reduced ability to localize the active compound in the bilayer, resulting in less efficient encapsulation or active compound loading.

For improving the efficiency of encapsulation of CBD into liposomes, novel lipid vesicular carriers containing a relatively high percentage of ethanol (ethosomal carrier) were designed [25]. Used as nanocarriers, they were especially designed for the efficient delivery of therapeutic agents with different physicochemical properties into deep skin layers and across the skin [26]. Such ethosomal carriers have been used to increase the efficiency of CBD encapsulation, resulting in an efficient anti-inflammatory system to be

used in rheumatic diseases [27]. Further, M. Çağdaş and co-workers [28] reported the nanostructured lipid carrier (NLC) resulted in good protection of entrapped THC and CBD, thus enabling the slow release of active substances. Nanostructured lipid carriers and solid lipid nanoparticles (SLNs) are non-biotoxic since they are biodegradable. Besides, they are highly stable [29]. Micro- and nano-emulsions of active cannabis ingredients (cannabinoids and terpenes) have also been presented in a patent [30] that proposed rectal–vaginal and solid oral dosage forms.

Modern lifestyles dictate the increasing use of tampons, which opens up new possibilities for their use in gynecology. Tampons today not only provide protection during the menstrual cycle, but also provide protection against physiological or pathological vaginal discharge and may also serve as a drug delivery system. Therefore, hypothetically, if CBD is placed on the tampon, when in use it will release the active substance directly into the uterus, which is the source of pain. Moreover, the smaller amount of CBD can be used for the same effect, compared to oral use. On the topic of CBD use for functionalizing the tampon, only a few technologies have been developed, which leaves room for improvement. In the field of sanitary products, i.e., tampons, the company Foria [31] developed Foria Relief capsules, applicable for vagina use and local pain relief. However, Foria Relief has not been approved by the U.S. Food and Drug Administration, nor has it been clinically tested. Daye [32] is, beside many other inventors, also a company producing CBD-infused tampons; compared to other similar products, their product contains 30% of CBD. Moreover, the results of performed clinical testing are not yet known.

Thus, the aim of this study was to use the nano/microencapsulation process in order to embed the CBD into liposomes as a potential coating for cellulose sanitary material. When it is encapsulated in the liposome and placed/crosslinked onto the tampon, it can, through the slow-release mechanism, release CBD directly into the uterus and relieve the pain significantly. The liquid formulation was characterized using the particle size, polydispersity, zeta potential, and encapsulation efficiency. The most optimal formulation was then used for coating, through dipping the cellulose fiber bands in the preparation of the functional tampon. The cellulose-encapsulated CBD composite was chemically analyzed using Fourier transform infrared spectroscopy (FTIR) and gravimetry, while the fiber morphology was evaluated using scanning electron microscopy (SEM). The bioactivity of the functionalized composite was evaluated by antioxidant and antimicrobial efficiency. The effect of CBD release was evaluated through desorption in water. Factors such as the increasing number of working women, the growing awareness of feminine hygiene, and increasing health problems such as skin irritation and rashes are also contributing to the growth of the market for biodegradable sanitary towels around the world. Thus, the environmental impact of the CBD composite tampon was assessed by biodegradability tests over a period of one month.

## 2. Materials and Methods

### 2.1. Materials Preparation

#### 2.1.1. Liquid Formulations

The liquid formulations were prepared according to procedure [33] using 3% CBD drops (trade name Be-Hempy, Toro Green d.o.o., Škofljica, Slovenia), Tryptone Soya Agar with Polysorbate 80 and Lecithin (0.7 g/L) from Sigma Aldrich (Vienna, Austria), edible refined sunflower oil (commercial oil Zvezda, Tovarna olja Gea, d.d., Slovenska Bistrica, Slovenia), deionized water and glycerol (purity $\geq$ 99.0%, Sigma Aldrich, Vienna, Austria).

A total of 10 g of lecithin and 50 g sunflower oil were mixed and stirred in a hot bath (T = 30 °C) until the lecithin was dissolved. Then slowly the known amount of the 3% CBD was added, followed by addition of deionized water and glycerol (0.5%). The prepared liquid formulation was stirred for 1 h, resulting in a homogeneous solution. The CBD liposome dispersion was kept under sonication (Sonics VCX-750 Vibra Cell, Newtown, CT, USA) for 7 min (1 s ON–1 s OFF, at T = 25 °C, frequency = 20 kHz at 80% power). The nano/micro CBD liposome were purged with the nitrogen gas for one hour to stabilize

them. The basic principle of the method used was derived from the nanoencapsulation of fish oil into liposomes [33].

The optimal ratio between mass of added sunflower oil and mass of added lecithin was 5:1, based on preliminary experiments.

Liquid formulation without CBD was prepared according to the abovementioned procedure, only without CBD, for testing of particle size, polydispersity index (PDI), and zeta potential.

### 2.1.2. Functionalized Cotton and Viscose Tampons

Basic tampon materials (in the form of bands) made from cellulose fibers (natural cotton from Tosama) and viscose fibers (Lenzing Fibers GmbH, Lenzing, Austria) were used for functionalization in this study. Both basic materials were kindly supplied by Tosama and were dermatologically tested and did not cause irritation or allergies. Both viscose fibers were pure and highly absorbent, specially designed for tampons.

The cotton and viscose tampon materials were dipped for 30 min into the prepared liquid formulation containing CBD encapsulated in liposomes, followed by removing the excess liquid using the Foulard squeezing rollers (Foulard, WERNER MATHIS AG, Oberhasli, Schweiz) to get 80% wetting. After that, the samples were air-dried. The fibrous sample description of functionalization is listed in Table 1.

**Table 1.** List of tampon samples and description of functionalized procedure used.

| Description of Sample |
| :---: |
| Cotton tampon, non-treated |
| Viscose tampon, non-treated |
| Cotton tampon, dipped into cannabidiol (CBD) liposome liquid formulation |
| Viscose tampon, dipped into CBD liposome liquid formulation |
| Cotton tampon, dipped into CBD oil only |

Reference materials coated only by CBD as control were also provided.

### 2.2. Methods for Characterization of Liquid Formulations

### 2.2.1. Particle Size, Polydispersity Index (PDI), and Zeta Potential Determination

Using the dynamic light scattering method (DLS) (Zetasizer Nano ZS, Malvern Instruments Ltd., Malvern, UK) [34], the particle size and polydispersity index (PDI) in each liquid formulation were determined. The liquid samples were injected into vials exposed to an electric field [35] in order to perform an electrophoresis experiment on the samples and to measure the velocity of the particles [36,37] using laser Doppler velocimetry (LDV) on a Zetasizer Nano ZS (Malvern Instruments Ltd., Malvern, UK). Each liquid sample was measured three times under the following conditions: T = 25 °C; t = 70 s; angle = 1730°, position = 0.45 mm.

### 2.2.2. Encapsulation Efficiency

The nano/micro CBD liposome dispersion was centrifuged (Hettich Zentrifugen, Universal 320R; using 5000 rpm; 15 min; 220 °C) in order to analyze the amount of non-encapsulated CBD in the supernatant. The amount of cannabidiol in the supernatant was determined by the gas chromatography–mass spectrometry (GC-MS, Perkin Elmer, Mass Spectrometer, Clarus SQ 8S) analytical method, kindly performed at the Institute for Chemistry, Ecology, Measurements and Analytics (IKEMA, Kidričevo, Slovenia). The encapsulation efficiency (EE%) was calculated by subtracting the free non-entrapped CBD substance concentration from the concentration of CBD substance added and dividing this by the concentration of total CBD substance added [38].

### 2.3. Methods for Characterization of Functionalized Tampons

#### 2.3.1. ATR FTIR Analysis

ATR FTIR spectra were recorded on a Perkin Elmer Spectrum GX spectrometer (Perkin Elmer FTIR, Omega, Ljubljana, Slovenia). The ATR accessory (supplied by Specac Ltd., Orpington, Kent, UK) contained a diamond crystal. A total of 16 scans were taken of each sample with a resolution of 4 cm$^{-1}$. All spectra were recorded at ambient temperature over a wavelength interval between 4000 and 650 cm$^{-1}$. Treated cotton or viscose tampons were analyzed in five different positions on the outer and inner sides to avoid the possible lack of homogeneity of the coating surface.

#### 2.3.2. Moisture Content and Mass Determination

The fibrous tampon materials after functionalization were dried to absolute dry mass. Thus, the moisture content mass of fibrous tampons before and after dipping into liquid formulation was determined using a moisture analyzer (Mettler Toledo HB4, Ljubljana–Dobrunje, Slovenia). The results presented the average value of the two measurements performed.

#### 2.3.3. Scanning Electron Microscopy Evaluation

The scanning electron microscopy (SEM, FE-SEM Supra VP 35, C. Zeiss, Oberkochen, Germany) was used for scanning the surface morphology of fibrous tampon samples. The samples (0.5 cm × 0.5 cm) were put onto aluminum holders, fixed by carbon tape to provide conductivity. The microscope Carl Zeiss Supra 35VP with a voltage level of 1 kV and variable working distance, using a 20–30 μm opening, was used. The images were taken in eight different places on each sample using different magnifications.

#### 2.3.4. Antioxidant Activity

The antioxidant activity was measured by UV/VIS spectrophotometer (Cary 60, Agilent, Santa Clara, CA, USA). The 7 mM of 2,2′-azino-bis(3-ethylbenzothiazoline-6-sulfonic acid) (ABTS) (0.0384 g) and 2.75 mM $K_2S_2O_8$ (0.0066 g) were put into the vessel and mixed with 10 mL of Milli-Q water and stored in a refrigerator for at least 12 h. A total of 2 mL of ABTS was diluted using phosphate-buffered saline (Fisher Scientific UK Ltd., Loughborough, UK) in order to obtain a solution with neutral pH that could form a stable and colored radical cation with the tested sample [39]. The 0.1 g of sample was mixed with 3.9 mL of prepared ABTS. The absorbance was measured according to the Beer–Lambert law at a wavelength of 734 nm in intervals of 0 s, 15 min, and 60 min. The samples were measured in triplicate. The inhibition was calculated according to Equation (1) as:

$$\text{Inhibition} = ((A_{ABTS} - A_{sample})/A_{ABTS}) \cdot 100\% \tag{1}$$

where $A_{ABTS}$ is the absorbance of ABTS radical and $A_{sample}$ is the absorbance of measured sample.

#### 2.3.5. Antimicrobial Activity

The antimicrobial testing was performed at the National Laboratory of Health, Environment and Food (Maribor, Slovenia) according to standard ASTM E2149-13a. The test was performed using *Escherichia coli* and *Staphylococcus aureus*. An incubated test culture in a nutrient broth was diluted using a sterilized 0.3 mM phosphate buffer ($KH_2PO_4$; pH = 6.8) in order to give a final concentration of 1.5–3.0·105 colony-forming units (CFU)/mL. This solution was used as a working bacterial dilution. Each sample was cut into small pieces (1 cm × 1 cm) and transferred to a 250 mL Erlenmeyer flask containing 50 mL of the working bacterial dilution. All the flasks were loosely capped, placed in the incubator, and shaken for 1 h at 37 °C and 120 rpm using a Wrist Action incubator shaker. After a series of dilutions using the buffer solutions, 1 mL of the diluted solution was plated in nutrient agar. The inoculated plates were incubated at 37 °C for 24 h and the surviving cells counted.

The antimicrobial activity of the tested samples was expressed as a log reduction in which the mean log10 density of bacteria in the jar containing the test sample after incubation for 1 h or 24 h was subtracted from the mean log10 density of bacteria in the jar prior to the addition of the test sample [40].

### 2.3.6. Desorption Evaluation

The release kinetics of the active substances from the treated tampon materials were monitored by the ABTS method, which was also used to determine the antioxidant activity of the samples. A total of 0.5 g of each tampon sample (treated with encapsulated CBD, with CBD only, and untreated) was put in 25 mL of distilled water and stirred in a shaker for 8 h as the most prolonged time of tampon usage (pH to 4.5 adjusted by concentrated lactic acid). The samples were then carefully removed from the aqueous medium. A total of 3.9 mL of prepared ABTS was added to 1.5 mL of each desorption bath sample and measurements were performed at a wavelength of 734 nm. The initial absorbance of the ABTS mixture prepared was 0.7263 nm. If CBD desorption occurred, it reacted with ABTS and thus the initial concentration in such desorption baths decreased.

### 2.3.7. Biodegradability Test

A biodegradability test was performed in accordance with the standard Textiles-Determination of the resistance of cellulose-containing textiles to micro-organisms (soil burial test, part 1: Assessment of rot-retardant Finishing (ISO 11721-1:2001)). After one month of exposing samples to soil, as determined in the standard, the samples were analyzed gravimetrically as already explained above, and the classic spectrophotometric determination of methylene blue dye absorption for determining carboxylic groups was done.

## 3. Results and Discussion

### 3.1. Liquid Formulations

### 3.1.1. Particle Size, PDI, and Zeta Potential Determination

After visual observation, the liquid formulation with encapsulated CBD was heterogeneous and, despite the stabilization by the nitrogen purification process used, sedimentation was still present. The results of determined particle size, polydispersity index (PDI), and zeta potential are presented in Table 2. Three independent measurements for each sample were performed, and thus average values are presented in Table 2, and graphically in Figure 1.

**Table 2.** The particle size, polydispersity index (PDI), and zeta potential of liquid formulations without (reference) and with CBD.

| Sample | Particle Size (nm) | PDI | Zeta Potential (mV) |
|---|---|---|---|
| Liquid formulation with CBD liposome | 1910.3 ± 90.7 | 0.8 ± 0.1 | −5.0 ± 1.6 |
| Liquid formulation without CBD, liposome only | 1559.7 ± 142.0 | 0.7 ± 0.1 | 6.4 ± 0.4 |

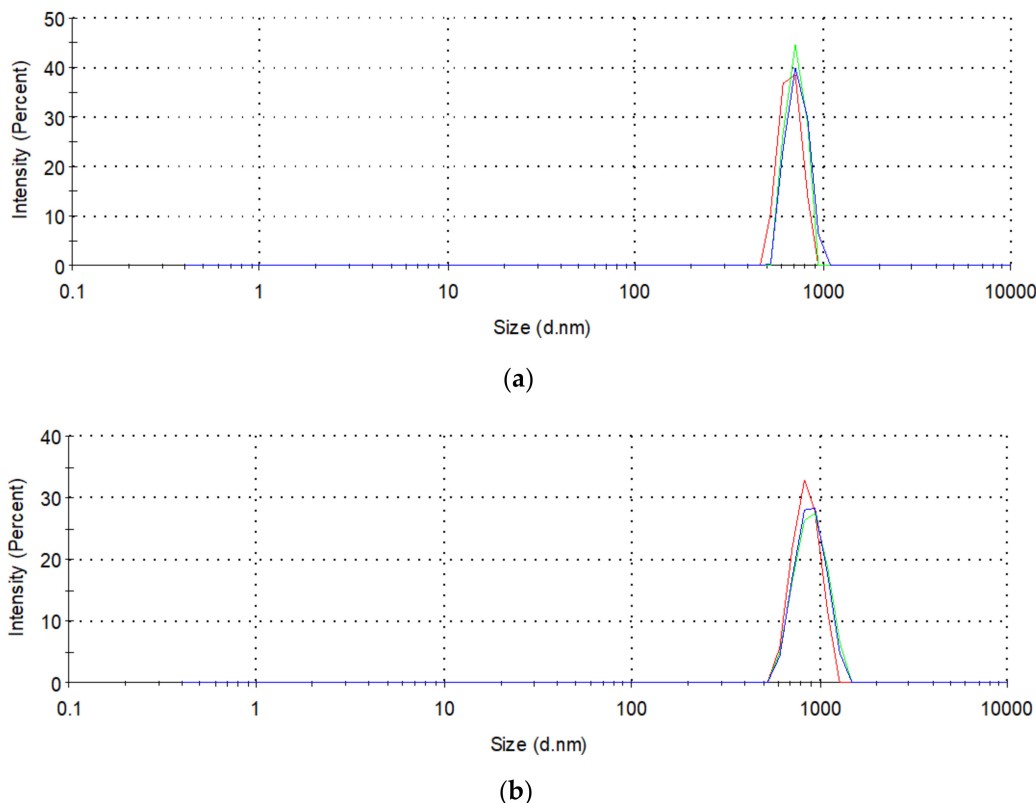

**Figure 1.** Dynamic light scattering method (DLS) size distribution of liquid formulation with CBD liposome (**a**) and liquid formulation without CBD, liposomes only (**b**).

The particle size in the liquid formulation containing CBD liposome was increased (to 19%) compared to the liquid formulation without CBD (liposomes only). In both formulations the synthesized particles were in micro scale. The higher polydispersity indexes in both cases indicated the presence of particles, showing different variations in their size, expressed more in the liquid formulation with CBD liposome compared to the sample containing liposomes only. The zeta potential (below 20 mV) indicated a non-stable dispersion and a tendency to agglomeration, regardless of whether the CBD was encapsulated in liposomes or not. Therefore, an intensive mixing of the formulation was required before application on cellulosic material.

### 3.1.2. Encapsulation Efficiency

The amount of cannabidiol amounted to 2927.7 mg/kg, as determined in the supernatant, using the GC-MS analytical method. The liquid formulation used for tampon functionalization contained 20 drops of 3% CBD. According to the producer's note, one drop contained 1.5 mg of cannabidiol, thus 20 drops contained 30 mg of CBD. Since the concentration of CBD oil used was 3% and the mass of 20 drops was 1000 mg, the concentration of CBD used in oil was 30 mg (CBD)/g, or 30,000 mg (CBD)/kg. The encapsulation efficiency, as was calculated using the above quoted data, amounted to 90%. We can emphasize a rather high and successful encapsulation of CBD into liposomes.

### 3.2. Functionalized Tampons

The cotton and viscose fibrous tampons were functionalized using coatings of the liquid formulation containing either CBD liposome microcapsules or CBD oil only. The external coated site of the tampons was further analyzed by using ATR FTIR spectroscopy.

3.2.1. ATR FTIR Evaluation

The ATR FTIR spectra (4000–650 cm$^{-1}$) of the non-treated cotton/viscose tampon and cotton/viscose tampon treated with the CBD oil or the CBD liposome liquid formulation were recorded in order to prove the functionalization of the CBD oil or the CBD liposome liquid formulation on tampons.

Spectra of the non-treated cotton tampon (A), cotton tampon treated with CBD oil only (B and C), and CBD oil as reference (D) are shown in Figure 2.

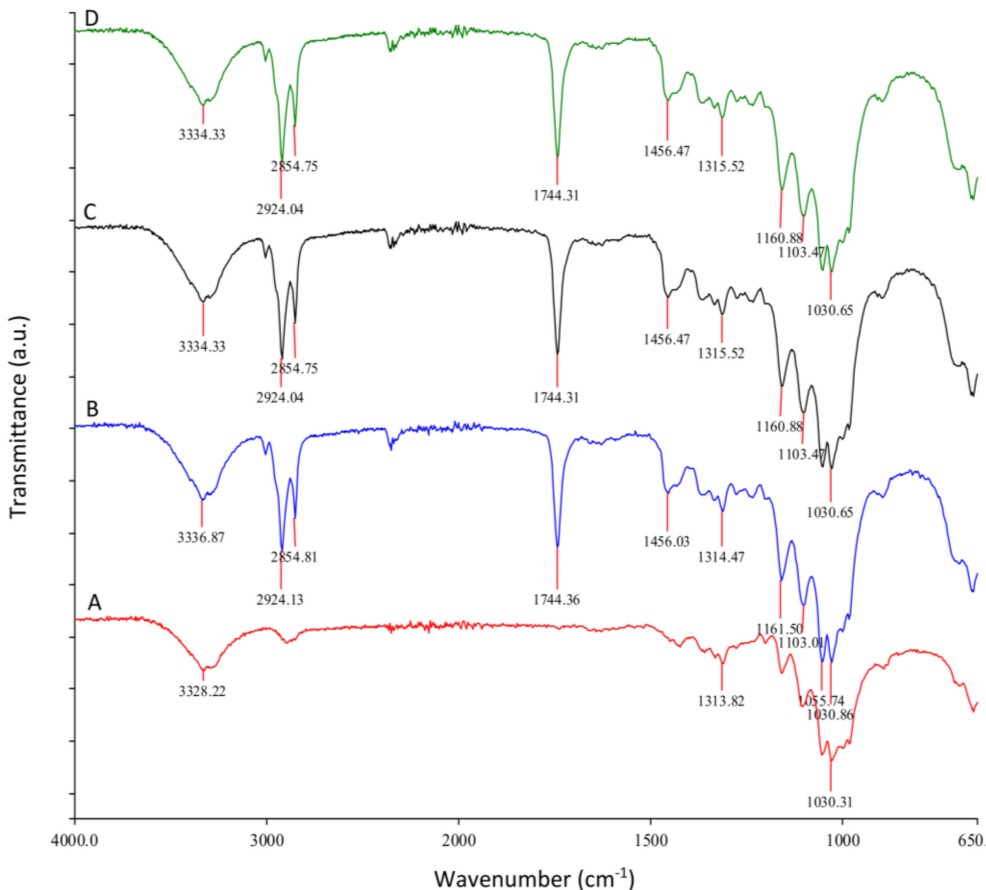

**Figure 2.** ATR FTIR spectrum of non-treated cotton tampon (**A** red); spectrum of cotton tampon treated with CBD oil only, recorded on outer side (**B** blue); spectrum of cotton tampon treated with CBD oil only, recorded on inner side (**C** black); and spectrum of CBD oil as reference (**D** green).

In the ATR FTIR spectrum of the non-treated cotton tampon (A) the typical bands for the cellulose are present. These bands are the broad band in the region around 3300 cm$^{-1}$ due to the OH stretching vibration, bands at around 2900 cm$^{-1}$ corresponding to the C–H stretching vibration, bands at 1313 cm$^{-1}$ and at 1030 cm$^{-1}$ assigned to CH$_2$ wagging vibration and C–O stretching vibration.

The changes of ATR FTIR bands, characteristic for the cellulose tampon, reflect the presence of the CBD oil on the cotton tampon. In the spectra of the cotton tampon treated with CBD oil (B and C), besides typical cellulose bands at 3300 cm$^{-1}$, 1313 cm$^{-1}$, and 1030 cm$^{-1}$, new bands appear; bands from 2900–2800 cm$^{-1}$ represent stretching vibrations of the methylene group, the band at 1744 cm$^{-1}$ comes from the ester carbonyl C=O functional group of triglycerides, and the band at 1456 cm$^{-1}$ corresponds to bending vibrations of the aliphatic groups. The bands in the region 1160–1100 cm$^{-1}$ are due to stretching vibrations of C–O ester groups (antisymmetric axial stretching and asymmetric axial stretching). The FTIR bands for the CBD oils matched the literature data [41]. The mentioned bands indicate the presence of CBD oil on the surface of the functionalized

cotton tampons; regardless of which side was measured, the bands' intensity from the outer side (spectrum B) and from the inner side (spectrum C) are almost identical.

Figure 3 summarizes the spectra of cotton tampons prepared by dipping into microcapsules of the CBD liposome liquid formulation (on the outer (spectrum B) and inner (spectrum C) sides of the tampons) along with the spectrum of non-treated cotton tampons (spectrum A). In the spectra of cotton tampons functionalized by the CBD liposome liquid formulation (spectra B and C), new bands at wave numbers from 2900–2800 cm$^{-1}$, 1744 cm$^{-1}$, 1456 cm$^{-1}$ and from 1160–1100 cm$^{-1}$ appear, which all belong to the CBD structure as pointed out above.

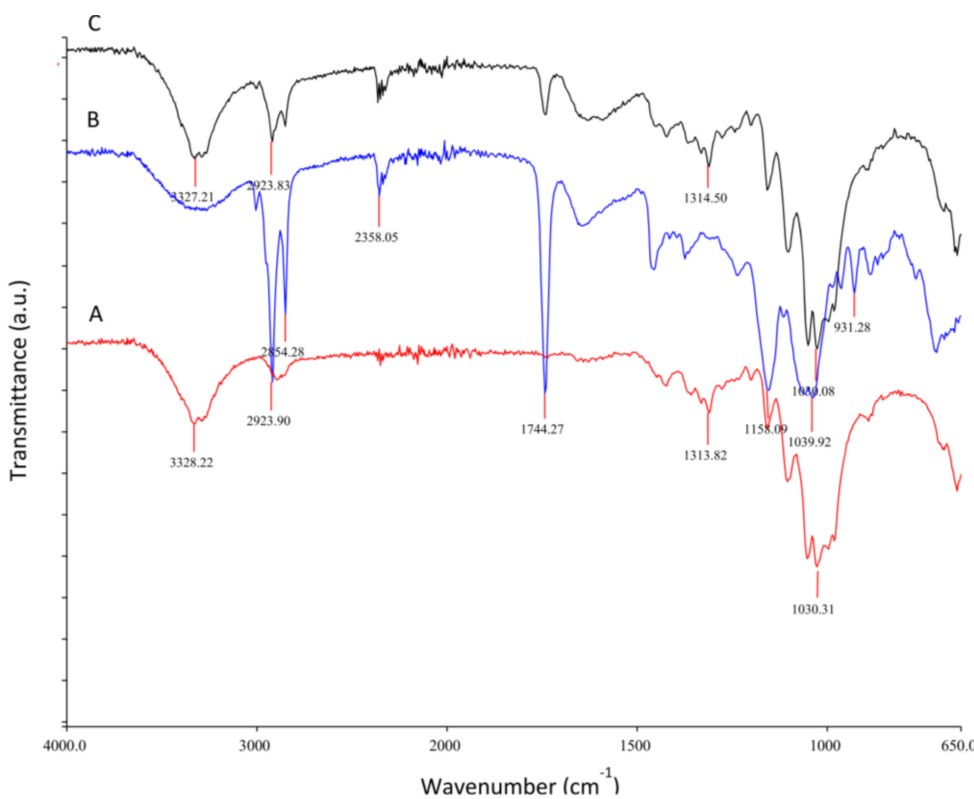

**Figure 3.** ATR FTIR spectrum of non-treated cotton tampon (**A** red), spectrum of CBD liposome functionalized cotton tampon on outer side (**B** blue), and spectrum of CBD liposome functionalized cotton tampon on inner side (**C** black).

We also noticed that there was no significant difference between the ATR FTIR spectra of the cotton tampon treated with CBD oil (spectrum B and spectrum C, Figure 2) and FTIR spectra of the cotton tampon functionalized by the CBD liposome liquid formulation (spectrum B and spectrum C, Figure 3). In spectra with the CBD liposome liquid formulation the following bands are present: bands in the region from 2900–2800 cm$^{-1}$ (stretching vibrations of the methylene group), two characteristic strong esters bands at 1744 cm$^{-1}$ and at 1160 cm$^{-1}$, and the band at 1456 cm$^{-1}$ (bending vibrations of the aliphatic groups). On the basis of this we can conclude that CBD was present on fiber surfaces.

Bands of the ATR FTIR spectrum of the cotton tampon functionalized by the CBD liposome liquid formulation on the outer side (spectrum B, Figure 3) are more intense than the bands of the ATR FTIR spectrum of the cotton tampon functionalized by the CBD liposome liquid formulation on the inner side (spectrum C, Figure 3). From this we can conclude that there was more CBD liposome liquid formulation coated on the outer side. The reason for this behavior is the heterogeneity of tampon mash.

Figure 4 compares the spectra of the viscose tampon prepared by dipping into the

CBD liposome liquid formulation (on the outer (spectrum B) and inner (spectrum C) sides of the tampon) with the spectrum of the non-treated viscose tampon (spectrum A).

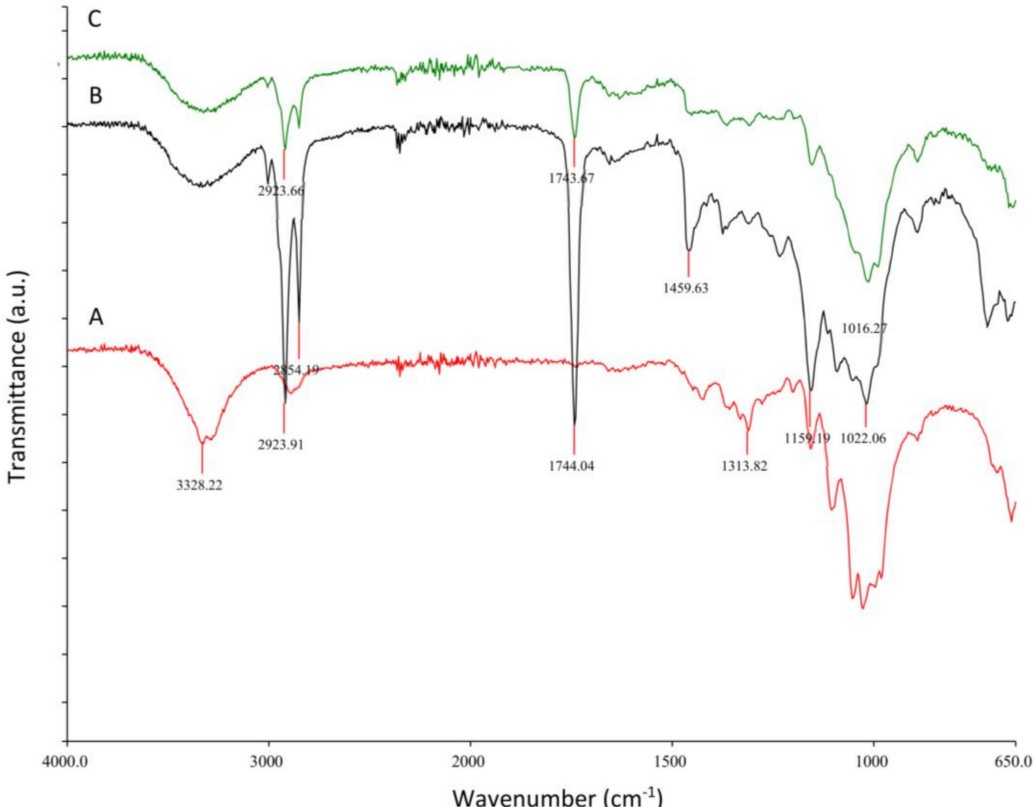

**Figure 4.** ATR FTIR spectrum of non-treated viscose tampon (**A** red), spectrum of CBD liposome functionalized viscose tampon on outer side (**B** black), and spectrum of CBD liposome functionalized viscose tampon on inner side (**C** green).

Because cotton and viscose fibers consist of the same material, mainly cellulose, it was not surprising that the ATR FTIR spectrum of the non-treated cotton tampon (spectrum A, Figure 2) and the ATR FTIR spectrum of the non-treated viscose tampon (spectrum A, Figure 4) had the same signals.

There was no significant difference between the ATR FTIR spectra of cotton tampons functionalized by the CBD liposome liquid formulation (spectrum B and spectrum C, Figure 3) and the ATR FTIR spectra of viscose tampons functionalized by the CBD liposome liquid formulation (spectrum B and spectrum C, Figure 4). In all spectra the following bands are present: stretching vibrations of the methylene group in the area from 2900–2800 cm$^{-1}$, two characteristic strong esters bands at 1744 cm$^{-1}$ and at 1160 cm$^{-1}$, and bending vibrations of the aliphatic groups at 1456 cm$^{-1}$.

Bands of the ATR FTIR spectrum of the viscose tampon functionalized by the CBD liposome liquid formulation on the outer side (spectrum B, Figure 4) have higher intensity than the bands of the ATR FTIR spectrum of the viscose tampon functionalized by the CBD liposome liquid formulation on the inner side (spectrum C, Figure 4), indicating more CBD liposome liquid formulation was deposited onto the outer side. The differences in the intensities of CBD liposome on cotton tampons compared to viscose tampons could be due to the different lengths of tampon fibers, namely, cotton fibers are longer compared to more fine and short viscose fibers. In non-woven tampon mash with shorter fibers, the liquid formulation of microcapsules penetrated perpendicular to the length of the tampon easier than in a more core mash.

### 3.2.2. Moisture Content and Mass Determination

In Table 3 mass before and after functionalization using the CBD liposome liquid formulation and CBD oil only (dry mass) is presented.

**Table 3.** Mass of original (non-dried) and dry mass of cotton and viscose tampon samples before and after functionalization with CBD liposome liquid formulation and CBD oil.

| Sample | Mass of Non-Treated Sample (g) | | Mass of Functionalized Sample (g) | | Δ Mass (g) |
|---|---|---|---|---|---|
| | Original | Dry | Original | Dry | |
| Cotton CBD liposome | 2.8 ± 0.2 | 2.7 ± 0.1 | 6.5 ± 0.3 | 5.9 ± 0.4 | 3.2 ± 0.5 |
| Viscose CBD liposome | 2.3 ± 0.0 | 2.1 ± 0.0 | 5.7 ± 0.2 | 5.2 ± 0.2 | 3.1 ± 0.2 |
| Cotton CBD oil | 2.8 ± 0.0 | 2.6 ± 0.4 | 3.8 ± 0.0 | 3.6 ± 0.3 | 1.0 ± 0.0 |
| Viscose CBD oil | 2.3 ± 0.0 | 2.0 ± 0.1 | 2.8 ± 0.1 | 2.4 ± 0.2 | 0.4 ± 0.0 |

The results in Table 3 point out that approximately 3 g of the CBD liposome liquid formulation was coated onto the cotton and viscose fibrous tampon samples. Thus, the increase in final fiber mass after the CBD liposome functionalization was 121.7% ± 26.8% for cotton and 148.0% ± 9.5% for the viscose sample. The high standard deviation referred to non-uniform coating. As the pure CBD oil coating was much less viscous and had a lower molecular weight, only a 37.5% increase in final mass was observed after coatings onto cotton, whereas only a 20% increase in mass was observed with coatings onto viscose.

### 3.2.3. Scanning Electron Microscopy Evaluation

The imaging of the microstructure and morphology of the untreated and CBD liposome functionalized cotton and viscose fibrous tampons, and the cotton sample functionalized using only CBD oil, are presented in Figure 5.

**Cotton fibrous sample**

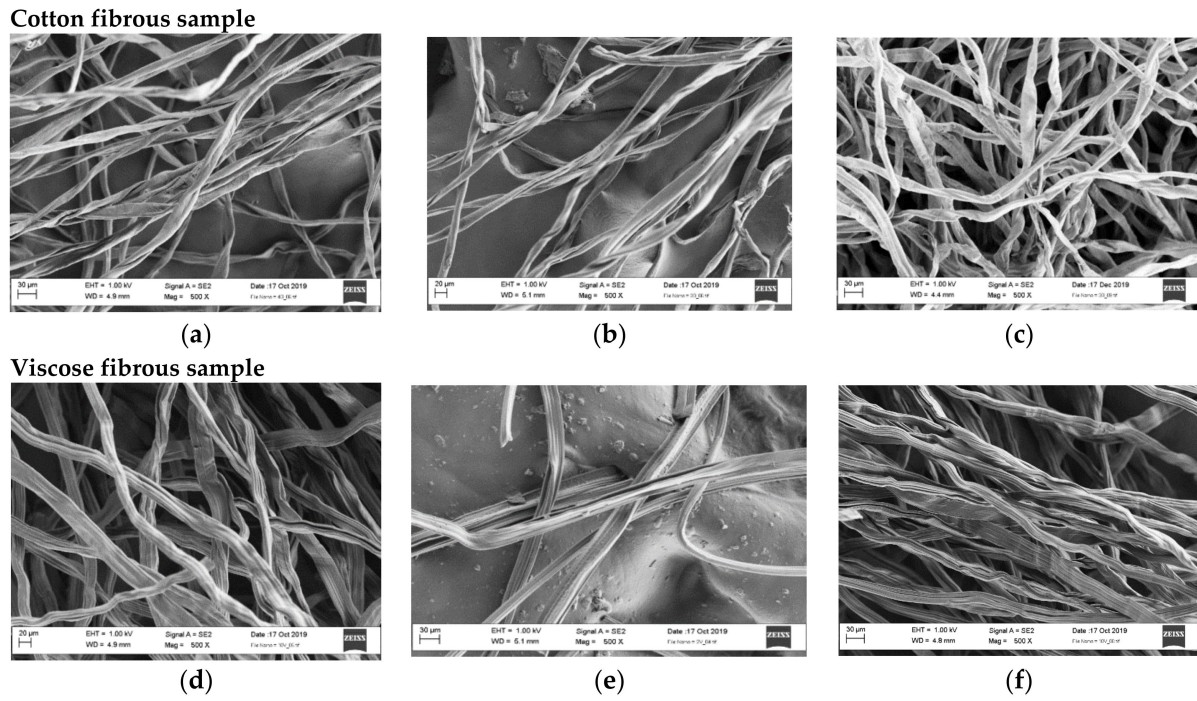

**Viscose fibrous sample**

**Figure 5.** The SEM images of (**a**) untreated cotton fibrous tampon, (**b**) cotton fibrous sample functionalized by CBD liposome liquid formulation, (**c**) cotton fibrous sample functionalized by CBD oil, (**d**) untreated viscose fibrous sample, (**e**) viscose fibrous sample functionalized by CBD liposome liquid formulation and (**f**) viscose fibrous sample functionalized by CBD oil at magnification factor range of 500×.

The pure cotton fibers are shown in Figure 5a, indicating the diameter range between 20 µm and 25 µm. The structure is not homogenous, since between the fibers different sizes of particles ranging from 20 to 200 µm can be seen. Figure 5b shows the structure of the cotton fibrous sample functionalized by the CBD liposome liquid formulation. The image shows fibers that interweave in a three-dimensional manner with the same diameter as the untreated sample. The particles indicate the presence of the liposomes containing CBD, and since they are distributed between the fibers, one can claim that the liquid formulation was absorbed into the fibrous tampon sample. Figure 4c presents the image of CBD oil functionalized cotton fibrous sample. In some places the fibers are glued together. The presence of oil drops cannot be seen; thus, one can assume that the drops were absorbed into the fibrous structure, causing an adhesion effect. The latter was expected because CBD alone is a much smaller molecule compared to the CBD liposome as macromolecule. The fibers have the same diameter as the untreated one, but are more roundish.

In Figure 5d the untreated viscose sample is shown, and in Figure 5e, the SEM micrograph of the viscose fibrous sample functionalized by the CBD liposome liquid formulation can be seen. In the latter image, one can see the occurrence of small particles, indicating the presence of CBD liposome. The clear image of the viscose CBD liposome functionalized sample shows that fibers are shorter, more rarely distributed, and interrupted here and there. The viscose fibers, untreated and functionalized, appear as the same diameter as the cotton fibers (i.e., 20 µm). SEM spectra for all functionalized samples, due to the presence of particles of different shapes, prove that CBD liposome are attached onto the fibers. Figure 5f shows the SEM image of CBD oil on viscose. There are no major noticeable differences between the SEM image of CBD oil on the viscose tampon (Figure 5f) and the SEM reference image viscose tampon (Figure 5d). The only difference is that the fibers prepared by CBD oil coatings on viscose pads are denser altogether. Apparently, due to the lower application of CBD oil on the viscose tampons (according to gravimetric analysis) compared to the cotton tampons, it is not possible to find significant differences. It is concluded that CBD oil was absorbed into the fibrous structures.

### 3.2.4. Antioxidant Activity

The oxidant inhibition (in %) of cotton and viscose fibrous tampons, untreated, functionalized by CBD liposome liquid formulation and cotton fibrous tampons functionalized only by CBD oil drops, is presented in Figure 6, as a function of time.

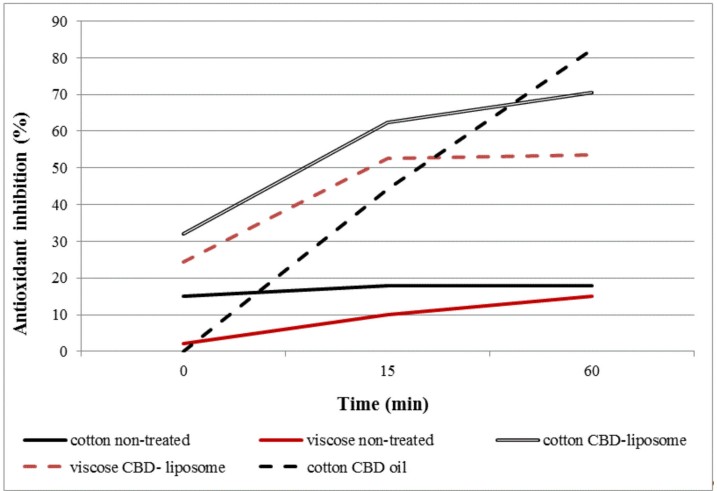

**Figure 6.** The antioxidant capacity (as inhibition %) of non-treated, CBD liposome, and CBD oil only functionalized cotton samples compared to non-treated and CBD liposome functionalized viscose samples.

The untreated cotton and viscose samples showed low inhibition of antioxidant activities, which were generally less than 20%. The curve of the cotton sample, functionalized by CBD oil only, shows the steepest incline, resulting in the highest oxidant inhibition (i.e., 82.3%) after 60 min of monitoring, indicating that high antioxidant capacity can be preserved even longer than 60 min. On other hand this can indicate that non-encapsulated CBD evaporated from the tampon rather fast. In addition, the tampon mash (regardless if cotton or viscose) functionalized by the CBD liposome liquid formulation, indicated an effective encapsulation with a slow or even tailored release of CBD. The antioxidant capacity of the cotton CBD liposome functionalized sample after 15 min amounted to 62.4% and, with a prolonged release time (60 min), up to 70.6%. The indicated release process of the viscose CBD liposome functionalized samples, where the curve is less steep, showed a lower release with this time interval, while the inhibition after 60 min remained the same as after 15 min and amounted to 53%. It appears that an equilibrium release was achieved after 15 min.

Results of antioxidant activity show that the CBD liposome formulation on tampons inhibits the inflammation through radical inhibition [42]. Further, it can be concluded that, compared to CBD-treated tampons, the CBD liposome functionalized tampon released CBD in a more controlled manner. It is proven that the antioxidant activity can be tailored by the amount of added CBD liposome.

### 3.2.5. Antimicrobial Activity

The antimicrobial testing was performed using *E. coli* DSM 1576 and *S. aureus* DSM 799. For the antimicrobial test, we selected the most optimal sample. From our point of view, this was cotton. It had slightly better physical–chemical properties after functionalization. In addition, cotton is more used and appreciated in the medical textile field due to its better biodegradability. It is more environmentally and human friendly, as fewer chemicals are used in its production. The results of antimicrobial activity obtained by cotton samples dipped into the CBD liposome liquid formulation and cotton samples functionalized only with CBD oil, compared to non-treated samples, are presented in Table 4. The results are presented as reduction (%) and as log reduction.

**Table 4.** Antimicrobial activity of an untreated cotton fibrous sample, a cotton fibrous sample functionalized by CBD liposome liquid formulation, and a cotton fibrous sample functionalized only by CBD oil.

| Sample | $t_0 = 1$ h | $t_1 = 24$ h | Reduction (%) | Log Reduction |
|---|---|---|---|---|
| | | *E. coli* **as Test Organism** | | |
| Working culture | $1.6 \times 10^5$ | $1.8 \times 10^5$ | / | / |
| Cotton non-treated | / | $8.3 \times 10^5$ | 0 | 0 |
| Cotton CBD liposome | / | $1.3 \times 10^5$ | 30 | 0.15 |
| Cotton CBD oil | / | $1.6 \times 10^5$ | 13 | 0.06 |
| | | *S. aureus* **as test organism** | | |
| Working culture | $1.5 \times 10^5$ | $3.2 \times 10^4$ | / | / |
| Cotton non-treated | / | $1.1 \times 10^5$ | 0 | 0 |
| Cotton CBD liposome | / | $5.1 \times 10^3$ | 84.4 | 0.81 |
| Cotton CBD oil | / | $7.0 \times 10^3$ | 78 | 0.67 |

The non-treated cotton sample did not show antimicrobial activity on either of the used microorganisms. Both cotton functionalized samples with CBD only showed inhibition of both used bacteria, while the sample functionalized by CBD liposome capsules showed greater effect. The cotton CBD oil sample showed 13% reduction of Gram-negative bacteria

*E. coli* and 78% reduction of Gram-positive bacteria *S. aureus*. The cotton CBD liposome sample showed higher inhibition for both bacteria; i.e., 30% for *E. coli* and 84.4% for *S. aureus*.

The antimicrobial efficiency was greater in the case of *S. aureus* microorganisms, taking into account both used functional liquid formulations (i.e., CBD liposome and CBD oil). The higher resistance of Gram-positive bacteria than Gram-negative (G–) bacteria to CBD liposome and CB oil functionalization was possibly due to differential membrane structures of these bacteria. Gram-negative bacteria have an outer phospholipid membrane that acts as a barrier and renders the membrane impermeable to lipophilic constituents [43,44]. In Gram-positive bacteria, direct contact between hydrophobic components and the phospholipid layer of the cell membrane occurs due to lack of an outer phospholipid membrane. This results in increased ion permeability, deterioration of protein components such as enzymes, and excretion of the intracellular constituents [44,45].

Our results showed that liposomes and CBD act synergistically to slightly improve bacterial inhibition. This may also be connected to controlled release of CBD when embedded into liposomes. The use of CBD alone had a limited effect, while the use of CBD encapsulated in liposomes showed a greater reduction in both bacteria, although *S. aureus* was more susceptible. Apart from the loading effect offered by liposomes, another advantage of lipid-based antimicrobial delivery systems is their fusogenicity, i.e., the ability of liposomes to fuse with the outer membrane of bacteria, due to the fluidity of their phospholipid bilayer structure [46]. The liposomal phospholipid bilayer resembles the structure of bacterial cell membranes, which facilitates fusion based on similarity. Upon fusion, high antimicrobial doses are directly available inside a bacterium [47,48], since the liposome nano/micro particles possess a great specific area, on which part the CBD can be deposited and thus in a greater amount also released.

### 3.2.6. Desorption Evaluation

The results of desorption evaluation, using antioxidant measurements and conductivity determination, are presented in Table 5.

**Table 5.** The desorption results of non-treated, CBD liposome, and CBD oil functionalized cotton and viscose fibrous samples, obtained by 2,2′-azino-bis(3-ethylbenzothiazoline-6-sulfonic acid (ABTS) inhibition and electrical conductivity determination.

| Sample | Desorption Determination (4 h) | ΔA (in %) | Desorption Determination (8 h) | ΔA (in %) |
|---|---|---|---|---|
| | Absorbance (A) | – | Absorbance (A) | – |
| PBS buffer + ABTS | 0.7263 | – | 0.7263 | – |
| viscose non-treated | 0.7200 | −0.9 | 0.7150 | −2 |
| cotton non-treated | 0.7188 | −1 | 0.7277 | +0.1 |
| cotton CBD liposome | 0.6329 | −13 | 0.5244 | −28 |
| viscose CBD liposome | 0.6812 | −6 | 0.5143 | −29 |
| cotton CBD oil | 0.7000 | −3 | 0.7002 | −3.6 |

It was assumed that if CBD was released from tampons it would react with ABTS radical and decrease its concentration. The initial absorbance of the PBS buffer with ABTS radical was 0.7263. The absorbance of the supernatant by the cotton and viscose fibrous mash sample remained almost unchanged at both desorption times (4 and 8 h), compared to the initial absorbance (i.e., PBS buffer with ABTS), while for the sample functionalized by CBD oil, the absorbance only slightly changed at both times. This could be explained by CBD insolubility in water. The absorbance of the supernatant by the cotton and viscose fibrous sample functionalized by the CBD liposome liquid formulation was 13% and 6% lower at a desorption time of 4 h, compared to the initial ABTS solution. This may indicate a slight CBD desorption process, as the cationic radical ABTS reacted with an antioxidant CBD substance released from the CBD liposome capsules. With a prolonged desorption

time of 8 h the decrease of absorbance and consequent desorption of CBD increased. The differences for both samples amounted to around −30%, which suggests that a certain amount of CBD was released in the simulation of the acidic environment (as in the real case of the vagina).

It has to be pointed out that the desorption process using the oil medium as a bath should be also carried out in the future in order to analyze the supernatant of the fibrous sample functionalized by CBD oil only. However, in our opinion the acidic aqua environment was nearer to the real vagina environment than pure oil conditions, which is why the desorption experiment was performed in this way.

Obviously, a small release of CBD by these functionalized tampons was expected, due to its good antioxidant and antimicrobial activity. Moreover, the remaining CBD may also have an active role as a surface-bound bioactivator of the tampon at the interface with physiological media (vaginal mucosa) and in this way takes care of the healthy environment of the vagina and overcomes pathogenic processes.

### 3.2.7. Biodegradability

Biodegradable polymers degrade due to microorganisms and/or enzymes, and the rate can take hours to years, according to the nature of polymers [49]. Biodegradability was tested through mass loss of samples after one month exposure to soil under the conditions within the required standard. Figure 7 presents visual observations of cotton tampons functionalized with the CBD liposome liquid formulation after the soil burial test. Samples were taken after each week for a period of one month. The results show that 63% of non-treated cotton samples and 56% of non-treated viscose samples decomposed. In cotton and viscose samples functionalized using the CBD liposome liquid formulation, 47% and 44% of mass losses, respectively, were observed. Coating of cellulose fibers postponed cellulose degradation due to the presence of oil and natural surfactant. According to Trofin et al. [50] biodegradation of CBD in a one-year test was about 10% in the case where the samples were stored in the darkness at 4 °C, and about 19% in the case where the samples were stored in the laboratory light at 22 °C. Different lipids exposed to soil biodegraded in few months [51].

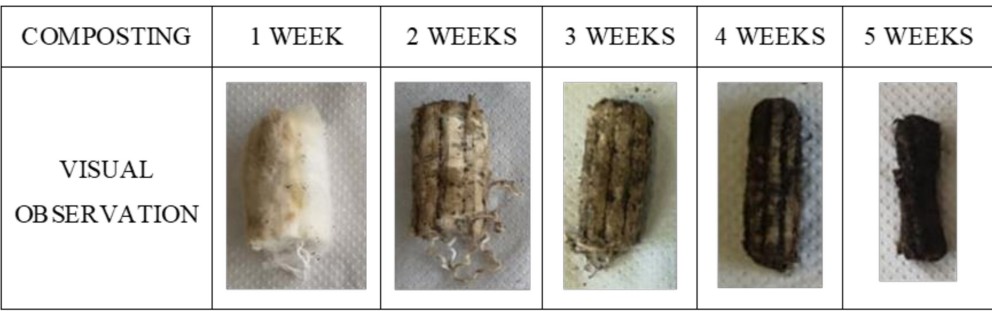

**Figure 7.** Cotton tampons functionalized using CBD liposome liquid formulation after soil burial test, visual observation after 1–5 weeks.

Using the methylene blue method, the carboxylic groups formed during the biodegradation of cellulose were calculated as mmol COOH/kg of the absolute dry sample. For both functionalized samples, a similar amount of newly introduced carboxylic group (increased in comparison to samples that were not subjected to soils) was detected; i.e., for the cotton CBD liposome sample the increase of carboxylic groups was the consequence of chemical degradation, which amounted to 15.1 mmol COOH/kg, and for the viscose CBD liposome sample 19.3 mmol COOH/kg. For non-treated ones, the higher amount of newly presented COOH groups was observed (36.5 mmol/kg for viscose and 28.2 mmol/kg for cotton) as the consequence of a more intensive process of biodegradability, as there was no addition of oils and surfactants.

Both methods clearly showed that viscose samples degraded more intensively than cotton. This is consistent with the fact that the polymer system of viscose has a much lower degree of crystallinity (35–40%) than cotton (65–70%), resulting in a weaker fiber that is more exposed to degradation. Moreover, the degree of polymerization of viscose is much lower than that of cotton [51]. Furthermore, the biodegradability of both reference samples was higher compared to samples coated with the CBD liposome liquid formulations. However, due to the loss of mass after one month and the external appearance of all fibers, one can conclude that the biodegradability was very successful and thus the samples expressed positive effects on the environment.

It must be pointed out that the functionalization process is technologically and economically suitable to be transferred into real-life conditions, while some process design and cost aspects must still be studied and respected in detail [52].

### 4. Conclusions

The CBD was encapsulated in liposome particles with an encapsulation efficiency of 90% and further used as coatings to functionalize the cotton and viscose tampons. The zeta potential and the PDI results indicated an unstable dispersion with broad micron particle distribution. The ATR FTIR results confirmed the presence of cannabidiol substances in the functional cellulose tampons. The scanning electron microscope images showed the presence of particles between the fibers of the cotton and viscose tampon samples functionalized by CBD liposome, whereas when only CBD oil was used, the drops of oil were absorbed into the fibrous structures.

CBD liposome functionalized tampons have both antioxidant and antimicrobial properties. The functionalized tampons exhibit antimicrobial properties that are more pronounced against Gram-positive bacteria, with the outer layer of the bacteria being less resistant.

Biodegradability compared to reference values was higher than expected, regardless of the essential oils present in CBD. This indicates the possibility for the development of an environmentally friendly biodegradable sanitary material. The use of hazardous chemicals and raw materials in ordinary sanitary towels has increased the interest of the female population in using biodegradable sanitary towels. The growing awareness of environmental protection is also driving the use of biodegradable raw materials in sanitary towels and medical textiles in these companies worldwide.

The CBD microencapsulation in liposomes showed a potential coating formulation for the development of an advanced biodegradable hygiene product for overcoming menstrual problems in daily use and as a curative treatment. It may also serve as a platform for other new sanitary and medical products.

**Author Contributions:** Conceptualization, L.F.Z. and J.V.V.; methodology, L.F.Z., Z.P., K.V., J.V.V., B.V., and L.T.; formal analysis, K.V. and J.V.V.; investigation, L.F.Z., J.V.V., and L.T.; data curation, K.V., J.V.V., and L.T.; writing—original draft preparation, L.F.Z., Z.P., and J.V.V.; writing—review and editing, L.F.Z., J.V.V., Z.P., K.V., L.T., and B.V.; and supervision, L.F.Z. and J.V.V. All authors have read and agreed to the published version of the manuscript.

**Funding:** Funding through the Research Programs P2-0118 and P2-0346 of the Slovenian Research Agency (ARRS) is gratefully acknowledged.

**Acknowledgments:** The authors acknowledge Nataša Poklar Ulrich for her support by preparing liquid formulations and Olivija Plohl for providing the scanning electron microscopy performance.

**Conflicts of Interest:** The authors declare no conflict of interest.

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
