# Peer review of "Microencapsulation of Cannabidiol in Liposomes as Coating for Cellulose for Potential Advanced Sanitary Material"

_coatings, doi:10.3390/coatings11010003_

Round 1

Reviewer 1 Report

  1. Explain the difference between the results of zeta potentials in Table 2. Please provide additional explanation why is the value of CBD-liposomes treated sample negative.
  2. Figure 4 and Table 3 do not provide information on CBD-oil treatment on viscose. Please add those results or explain expected trends.
  3. Since all treatments have been performed both on cotton and viscose material, please explain why the results of antimicrobial treatments have not been presented for the viscose fiber.

Author Response

Response to Reviewer 1 Comments

We thank to the reviewer for the review and for the provided comments. Comments were taken under consideration and the responses are given below.

Point 1: Explain the difference between the results of zeta potentials in Table 2. Please provide additional explanation why is the value of CBD-liposomes treated sample negative.

Response 1:

It has been previously shown in the literature (Ren, 2019) and (Villasmil-Sánchez, 2010) that liposomes show various surface charges (positive, negative, slight positive, and slight negative) depending on particle sizes and conformation. With addition of CBD into liposomes the size of particle increase slightly which prove the encapsulation of CBD. The changes in zeta potential values of both samples exhibited a slight difference between their zeta potential values, which suggests that the charge on the surface remains essentially the same after encapsulation. The charge reversal may indicate the presence of CBD also on the particle surface. However, due to very low values of zeta potential it may be seen that both particles are not stable and tend to agglomerate.

Ren, Hongwei, Yuwei He, Jianming Liang, Zhekang Cheng, Meng Zhang, Ying Zhu, Chao Hong, Jing Qin, Xinchun Xu, and Jianxin Wang. "Role of Liposome Size, Surface Charge, and Pegylation on Rheumatoid Arthritis Targeting Therapy." ACS Applied Materials & Interfaces 11, no. 22 (2019): 20304-15.

Villasmil-Sánchez, S., W. Drhimeur, S. C. Ospino, A. M. Rabasco Alvarez, and M. L. González-Rodríguez. "Positively and Negatively Charged Liposomes as Carriers for Transdermal Delivery of Sumatriptan: In Vitro Characterization." Drug Dev Ind Pharm 36, no. 6 (2010): 666-75.

Point 2: Figure 4 and Table 3 do not provide information on CBD-oil treatment on viscose. Please add those results or explain expected trends.

Response 2:

Thank you for the comments. We would like to apologize for the error as we apparently forgot the intake of the viscose-CBD sample. So we added the mass of non-treated sample and the mass of functionalized viscose tampon with CBD-oil on in Table 3.

In Figure 4 (Figure 5 in new manuscript) the SEM image of CBD-oil on viscose tampon was added. There are no major noticeable differences between SEM image of CBD-oil on viscose tampon and the SEM image of reference - viscose tampon. The only difference is that the fibres in CBD-oil on viscose tampon are denser together. Apparently due to the lower application of CBD oil (gravimetric analysis), compared to cotton, it is not possible to notice significant differences. For this reason, we have also not included a picture in the original version of the article.

Point 3: Since all treatments have been performed both on cotton and viscose material, please explain why the results of antimicrobial treatments have not been presented for the viscose fiber.

Response 3:

For the antimicrobial test we choose the most optimal sample. From our point of view this is cotton tampon. It shows a bit better physic- chemical characteristics. Moreover, the cotton is more used and appreciated in segment of medical textiles due to its better biodegradability. It is more environmentally and human friendly due to lower amount of chemical present during production process.

Reviewer 2 Report

The manuscript is interesting, in my opinion, it can be considered for publication pending the revision based on the following:

  1. In the abstract add more concrete results.
  2. In introduction add more background about the work with up-to-date citations.
  3. Why cannabidiol was chosen? Add concrete proof of microencapsulation.
  4. What about the toxicity of cannabidiol? Is it safe to use? Explain.
  5. Improve the quality and resolution of Figures 1, 2, 3 & 5 for a better understanding of the reader.
  6. Add FTIR of cannabidiol as a reference.
  7. Add significant difference in statistical analysis in Tables 1 & 2.
  8. Why PDI is very high and zeta potential is too low? Explain. Authors are advised to add DLS results (Figure) in text.
  9. Add more evidence of encapsulation. Also, add visual image of all the formulations.
  10. Add one more antioxidant test (such as DPPH) for the understanding of the results.
  11. Show the image of biodegradability.
  12. The discussion could be more solid. Revise it very carefully.
  13. Conclusion should be short and to the point rather than a summary of all the findings.
  14. Also, carefully revise the manuscript for linguistic and typos errors.

Author Response

Response to Reviewer 2 Comments

We thank to the reviewer for the review and for the provided comments. Comments were taken under consideration and the responses are given below.

Point 1: In the abstract add more concrete results.

Response 1:

Abstract has been change and concrete results were included.

Point 2: In introduction add more background about the work with up-to-date citations.

Response 2:

The following up-to date citations have been added in the Introduction.

Bruni, N., C. Della Pepa, S. Oliaro-Bosso, E. Pessione, D. Gastaldi, and F. Dosio. "Cannabinoid Delivery Systems for Pain and Inflammation Treatment." Molecules 23, no. 10 (2018).

Couch, D. G., C. Tasker, E. Theophilidou, J. N. Lund, and S. E. O'Sullivan. "Cannabidiol and Palmitoylethanolamide Are Anti-Inflammatory in the Acutely Inflamed Human Colon." Clin Sci (Lond) 131, no. 21 (2017): 2611-26.

Pellati, F., V. Borgonetti, V. Brighenti, M. Biagi, S. Benvenuti, and L. Corsi. "Cannabis Sativa L. And Nonpsychoactive Cannabinoids: Their Chemistry and Role against Oxidative Stress, Inflammation, and Cancer." Biomed Res Int 2018 (2018): 1691428.

Zagzoog, Ayat, Kawthar Mohamed, Hye Kim, Eunhyun Kim, Connor Frank, Tallan Black, Pramodkumar Jadhav, Larry Holbrook, and Robert Laprairie. "In Vitro and in Vivo Pharmacological Activity of Minor Cannabinoids Isolated from Cannabis Sativa." Scientific Reports 10 (2020).

Nichol, Kathryn, Colin Stott, Nicholas Jones, Royston A. Gray, Michael Bazelot, and Benjamin J. Whalley. "The Proposed Multimodal Mechanism of Action of Cannabidiol (Cbd) in Epilepsy: Modulation of Intracellular Calcium and Adenosine-Mediated Signaling (P5.5-007)." Neurology 92, no. 15 Supplement (2019): P5.5-007.

Point 3: Why cannabidiol was chosen? Add concrete proof of microencapsulation.

Response 3:

Cannabidiol (CBD) was chosen because it has great potential as an analgesic, anti-epileptic, muscle relaxant, anxiolytic, neuroprotectant and antipsychotic, as well as anti-inflammatory and antioxidant and cancer related pains. In addition to all above CBD also has "relaxing and analgesic effects", i.e. it could induce the muscles in the abdominal cavity to relax and stop contracting so aggressively, which would lead to dysmenorrhea - pain relief, in view of that fact CBD has been placed on the tampon, CBD can be released directly in to the uterus, which is the source of pain, so the smaller amount of CBD could be used for the same effect compared to oral use. By encapsulating CBD in liposome and placed/crosslinked on to the tampon controlled release can be achieved.

Concrete evidence of microencapsulation is that the particle size in a liquid formulation containing CBD liposomes was increased by 19% compared to a liquid formulation without CBD (liposomes only). The successful encapsulation of CBD has been demonstrated using the analytical GC-MS method. We would like to emphasize that the method for liposome formulation was taken from the literature data, where Ghorbanzade and collaborators (Ghorbanzade, Tahere, Seid Jafari, Sahar Akhavan, and Roxana Hadavi. "Nano-Encapsulation from Fish Oil to Nano-Liposomes and its application in the enrichment of yogurt". Food Chemistry 216 (2016).), successfully encapsulated fish oil. This validated technique was adapted in our work.

Point 4: What about the toxicity of cannabidiol? Is it safe to use? Explain.

Response 4:

Cannabidiol (CBD) has multiple pharmacological actions, little is known about its safety and side effect profile in animals and humans. There is abundance of public comments on the therapeutic effects of cannabinoids indicating the fact there has only been a limited number of clinical studies on this topic due to the illegal status of cannabinoides in most countries. Bergamaschi, et.al reviewed in vivo and in vitro reports on CBD administration across a wide range of concentrations, based on reports retrieved from Web of Science, Scielo and Medline. It was reported that based on advances in CBD administration in humans, controlled use of CBD may be safe in humans and animals. CBD use also carries some risks and cause some side effects. In general, authors concluded that controlled administration of CBD may be safe in humans and animals. More recent study on adverse efficient and toxicity reported that CBD has proven therapeutic efficacy for serious conditions and is likely to be recommended off label by physicians for several conditions. However, adverse effects and potential drug-drug interactions must be taken into consideration by clinicians prior to recommending off-label CBD. (This part has also been added to Introduction)

Point 5: Improve the quality and resolution of Figures 1, 2, 3 & 5 for a better understanding of the reader.

Response 5:

Quality and resolution of Figures 1, 2, 3 and 5 (In new manuscript Figure 2, 3, 4 and 6) have been improved and add to new version of manuscript.

Point 6: Add FTIR of cannabidiol as a reference.

Response 6:

FTIR spectrum of cannabidiol oil as reference has been added to the Figure 2 (In new manuscript).

Point 7: Add significant difference in statistical analysis in Tables 1 & 2.

Response 7:

Three independent measurements were performed for each sample. The Malvern measures the time-dependent fluctuations of light scattered by the liposomes and uses it to calculate the average size and polydispersity of the liposomes. The average value of three measurements were pointed out and are presented in the Table 2. Standard deviations are taken into account.

Point 8: Why PDI is very high and zeta potential is too low? Explain. Authors are advised to add DLS results (Figure) in text.

Response 8:

The fact that the PDI is very high and the zeta potential is too low is a phenomenon that is quite often expected. The absolute value of the zeta potential below 20 mV shows a non - stable dispersion and the tendency of the particles to agglomerate. When particles are subjected to agglomeration, the distribution of particles by size in this process is very wide. The size - heterogeneity of particles is expectable by their physical non-stability. Thus PDI is high and in accordance with this. DLS results are graphically added as Figure 1.

Point 9: Add more evidence of encapsulation. Also, add visual image of all the formulations.

Response 9:

DLS Size distribution of Liquid formulation with CBD – liposomes and liquid formulation without CBD, liposomes only have been added in manuscript (Figure 1). The particle size in liquid formulation containing CBD - liposomes was increased for 19% compared to liquid formulation without CBD (liposomes only). More comments are added in response 3.

Point 10: Add one more antioxidant test (such as DPPH) for the understanding of the results.

Response 10:

During the research only ABTS antioxidant test is used which is available in our laboratory. In all our previously research we perform ABTS test and test provide accurate information to understand the results. We could say that ABTS analysis is more suitable for our samples as it is performed in buffer solution, while DPPH analysis is performed in methanol. Since ABTS analysis is performed in buffer solution, this method is closer to the real conditions in the vagina. However, we thank the reviewer for the suggestion and will introduce this technique in the future for subsequent work.

Point 11: Show the image of biodegradability.

Response 11:

Figure 7 has been added to manuscript. Figure presented biodegradability, visual observation of cotton tampons functionalized with CBD - liposome liquid formulation after soil burial test. Samples were taken after each week for a period of 1 month.

Point 12: The discussion could be more solid. Revise it very carefully.

Response 12:

The discussion was revised in many parts to make it more solid. 

Point 13: Conclusion should be short and to the point rather than a summary of all the findings.

Response 13:

The conclusion has been rewritten.

Point 14: Also, carefully revise the manuscript for linguistic and typos errors.

Response 14: It has been again check for linguistic and typos errors.

Round 2

Reviewer 2 Report

The revision was appropriately made by considering the comments. No further comment.